# Analysis of virulence potential of *Escherichia coli* O145 isolated from cattle feces and hide samples based on whole genome sequencing

**Pragathi B. Shridhar**[1]**, Jay N. Worley**[2]**, Xin Gao**[2]**, Xun Yang**[2]**, Lance W. Noll**[3]**, Xiaorong Shi**[1]**, Jianfa Bai**[3]**, Jianghong Meng**[2]**, T. G. Nagaraja**[1]*

**1** Department of Diagnostic Medicine/Pathobiology, Kansas State University, Manhattan, Kansas, United States of America, **2** Joint Institute for Food Safety and Applied Nutrition and Department of Nutrition and Food Science, University of Maryland, College Park, Maryland, United States of America, **3** Veterinary Diagnostic Laboratory, Kansas State University, Manhattan, Kansas, United States of America

* tnagaraj@vet.k-state.edu

**Data Availability Statement:** Draft genome sequences of the 71 E. coli O145 strains are available in GenBank under bioproject accession no. PRJNA525675.

## Abstract

*Escherichia coli* O145 serogroup is one of the big six non-O157 Shiga toxin producing *E. coli* (STEC) that causes foodborne illnesses in the United States and other countries. Cattle are a major reservoir of STEC, which harbor them in their hindgut and shed in the feces. Cattle feces is the main source of hide and subsequent carcass contaminations during harvest leading to foodborne illnesses in humans. The objective of our study was to determine the virulence potential of STEC O145 strains isolated from cattle feces and hide samples. A total of 71 STEC O145 strains isolated from cattle feces (n = 16), hide (n = 53), and human clinical samples (n = 2) were used in the study. The strains were subjected to whole genome sequencing using Illumina MiSeq platform. The average draft genome size of the fecal, hide, and human clinical strains were 5.41, 5.28, and 5.29 Mb, respectively. The average number of genes associated with mobile genetic elements was 260, 238, and 259, in cattle fecal, hide, and human clinical strains, respectively. All strains belonged to O145:H28 serotype and carried *eae* subtype γ. Shiga toxin 1a was the most common Shiga toxin gene subtype among the strains, followed by *stx*2a and *stx*2c. The strains also carried genes encoding type III secretory system proteins, *nle*, and plasmid-encoded virulence genes. Phylogenetic analysis revealed clustering of cattle fecal strains separately from hide strains, and the human clinical strains were more closely related to the hide strains. All the strains belonged to sequence type (ST)-32. The virulence gene profile of STEC O145 strains isolated from cattle sources was similar to that of human clinical strains and were phylogenetically closely related to human clinical strains. The genetic analysis suggests the potential of cattle STEC O145 strains to cause human illnesses. Inclusion of more strains from cattle and their environment in the analysis will help in further elucidation of the genetic diversity and virulence potential of cattle O145 strains.

**Funding:** This material is based upon work that is supported by the National Institute of Food and Agriculture, U. S. Department of Agriculture, under award number 2012-68003-30155. The funder had no role in the study design, data collection and analyses, preparation of the manuscript or decision to publish.

**Competing interests:** The authors have declared that no competing interests exist.

## Introduction

In recent years, there is an increased incidence of non-O157 STEC-associated human illnesses. Six STEC serogroups, O26, O45, O103, O111, O121, and O145, are responsible for more than 70% of non-O157 STEC-associated human illnesses in the United States [1, 2]. In 2011, the US Department of Agriculture, Food Safety and Inspection Service declared these six non-O157 STEC as adulterants in ground beef and non-intact raw beef products [3]. The serogroup O145 is responsible for several outbreaks in the US and other countries, including Germany [4], Argentina [5] and Belgium [6]. In the US, two cases of *E. coli* O145 associated infection were reported in a day care in Minnesota in 1999 [7]. The serogroup was also responsible for a waterborne human illness in Oregon in 2005 [8], and in 2010, a multistate outbreak associated with the consumption of romaine lettuce, leading to 45% hospitalization, with 10% of the patients developing hemolytic uremic syndrome [9].

Cooper et al. (2014) analyzed the whole genome sequences (WGS) of two strains of *E. coli* O145:H28 that were associated with the romaine lettuce outbreak in the US and ice cream outbreak in Belgium, and compared them to genome sequences of *E. coli* and *Shigella*. They reported that O145 and O157 strains evolved from a common lineage, and the core genome profile of *E. coli* O145 strains was more similar to that of *E. coli* O157 than to other *E. coli* strains [10]. Carter et al (2016) studied the genetic diversity, population structure, virulence potential, and antimicrobial resistance profiles of environmental *E. coli* O145 strains (cattle feces, feral pigs, wild life, sediment, water and human clinical cases). They reported an extensive genetic diversity among the strains, and antimicrobial resistance appeared to be widespread in environmental strains with over half of the cattle strains resistant to at least one of the 14 antibiotics tested [11].

Cattle are major reservoirs of STEC, which harbor them in the hindgut and shed in the feces. Cattle feces are a main source of hide, and carcass contamination during harvest, potentially leading to foodborne illnesses in humans. Several studies have shown the association between prevalence of STEC in cattle feces and subsequent hide and carcass contaminations [12, 13]. However, it is essential to determine the virulence potential of the strains in order to estimate the risk associated with each source of contamination, and to design the intervention strategies to prevent foodborne illness in humans. The objective of our study was to assess the virulence potential of STEC O145 strains isolated from cattle feces and hide samples using WGS-based analysis. We have demonstrated the pathogenic potential of cattle O145 strains and their close similarity to human clinical O145 strains based on genetic characterization, suggesting the potential of cattle O145 strains to cause foodborne illness in humans.

## Materials and methods

The protocol was approved by the Institutional Animal Care and Use Committee (IACUC) for Kansas State University (IACUC # 3764).

### *Escherichia coli* O145 strains

A total of 69 STEC O145 strains isolated from cattle feces and hide samples, and two human clinical strains were used for whole genome sequencing. The strains isolated from cattle feces (n = 16) and hide swab samples (n = 53) were collected at feedlots and in abattoirs, respectively. The fecal strains were from two studies conducted in the summer months of 2013 [14] and 2014 [15]. The hide strains were from a study conducted in the summer months of 2015 and 2016 [16]. The details about sample collection, isolation and identification have been described previously [14–16]. Two human clinical strains obtained from Kansas Department of Health and Environment were included in the study. The strains were positive for *stx*1

(n = 62) only, *stx*2 only (n = 8), both *stx*1 and *stx*2 (n = 1), and *eae* (n = 71) by PCR [17]. The strains were cultured onto Tryptone Soy agar (TSA; BD Difco, Sparks, MD) slants and shipped overnight on ice to the University of Maryland for whole genome sequencing.

## DNA extraction and whole genome sequencing

*Escherichia coli* O145 strains on the TSA slants were restreaked onto blood agar and then subcultured in TSB. The genomic DNA was extracted from the broth culture using DNeasy Blood and Tissue Kit with the QIAcube robotic workstation (Qiagen, Germantown, MD). The genomic libraries were constructed using Nextera XT DNA Library Preparation Kit and MiSeq Reagent Kits v2 (500 Cycles) (Illumina, Inc.). Whole genome sequencing was performed using an Illumina MiSeq platform (Illumina, San Diego, CA). *De novo* genome assembly was performed using SPAdes 3.6.0 [18].

## Sequence analysis

The initial annotation of draft genomes of STEC O145 strains was performed using RAST (Rapid Annotation using Subsystem Technology; [19]). The O and H-types were identified using SerotypeFinder 1.1 (http://www.genomicepidemiology.org) and BLAST tools (https://blast.ncbi.nlm.nih.gov/Blast.cgi), respectively. The number of genes categorized as associated with virulence, disease and defense, mobile elements (phages, prophages, transposable elements, and plasmids), membrane transport, iron acquisition and metabolism, and stress response in each strain was determined using RAST. A analysis of variance test was performed to determine whether genome size, and number of genes associated with different functional categories were significantly different between O145 strains isolated from different sources. Tukey adjustment for multiple comparisons was performed, using SAS 9.4 with Proc Glimmix, to test each pairwise comparison for significant differences ($P < 0.01$), if the means were significantly different ($P < 0.01$). The virulence gene profile and antimicrobial resistance genes were determined using VirulenceFinder 1.4 [20] and ResFinder 2.1 [21], web-based tools developed by the Center for Genomic Epidemiology (CGE) at the Danish Technical University (DTU), Lyngby, Denmark (http://www.genomicepidemiology.org/). Plasmid and phage sequences were identified using PlasmidFinder v1.3 (https://cge.cbs.dtu.dk/services/PlasmidFinder/) and Phage Search Tool Enhanced Release (PHASTER; http://phaster.ca/), respectively. Clusters of regularly interspaced short palindromic repeats (CRISPR)-Cas system of *E. coli* O145 strains were characterized based on annotation by CRISPRone, a web-based tool (http://omics.informatics.indiana.edu/CRISPRone). The tool provides class, type, and subtype of CRISPR-Cas system and number, length and nucleotide sequences of repeats and spacers [22]. The sequence types (ST) of each strain were determined using *in silico* MLST tool, MLST v1.8 [23, 24], a web-based tool developed by CGE. The phylogenetic relationship among the STEC O145 strains of cattle and human origin was determined using Parsnp v1.2 (http://harvest.readthedocs.io/en/latest/content/parsnp.html) [25], which performs core genome alignment followed by construction of maximum likelihood tree. The tree was visualized using FigTree 1.4 software (http://tree.bio.ed.ac.uk/software/figtree/). *Escherichia coli* O145:H28 strain RM12581 (GenBank accession no. CP007136), isolated from romaine lettuce associated with multistate outbreak of STEC O145 infections in the United States (Cooper et al., 2014), were used as control for comparison.

## Nucleotide sequence accession numbers

Draft genome sequences of the 71 *E. coli* O145 strains are available in GenBank under bioproject accession no. PRJNA525675.

## Results

All the strains (n = 71) were confirmed to be of O145 serogroup by SerotypeFinder 1.1 using default parameters (select threshold for % ID = 85%, and select minimum length = 60%). All the STEC O145 strains carried *fliC*$_{H28}$. The flagellar genes of the strains showed $\geq$ 99% identity to the *fliC*$_{H28}$ reference sequence (GenBank accession no.LN555740, LN555741, LN649615).

### RAST subsystem summary

Based on the RAST subsystem annotation, the average draft genome size of STEC O145 strains isolated from cattle feces and hide were 5.41 (5.25–5.63) Mb and 5.28 (5.21–5.46) Mb, respectively. The average draft genome size of fecal strains was significantly larger ($P < 0.01$) than the hide strains. The average number of genes associated with mobile genetic elements (phages, prophages, transposable elements and plasmids) was significantly higher ($P < 0.01$) in strains isolated from cattle feces [260 (224–291)] compared to hide strains [238 (203–268)]. There was no significant difference in the average number of genes associated with membrane tranport, iron acquistion and metabolism, and stress response categories between O145 strains isolated from different sources. The average number of genes associated with the major subsystem categories in all the strains is provided in Table 1.

### Virulence genes

Of the 71 strains, 62 strains were positive for *stx*1 only (10 fecal, 51 hide and one human strains), seven strains for *stx*2 only (6 fecal and one hide strains) and one human strain was positive for both *stx*1 and *stx*2. Shiga toxin gene sequence was not identified in the genome of one of the STEC strain which was positive for *stx2* by end-point PCR. Shiga toxin 1a was the most common subtype in bovine fecal (10/16; 62.5%), hide (51/53; 96.2%) and human clinical strains (2/2; 100%). Shiga toxin 2a was present only in bovine fecal (5/16; 31.3%) and human clinical strains (1/2; 50%), but absent in cattle hide strains. Similarly, *stx*2c was present only in bovine fecal (1/16; 6.3%) and hide strains (1/53; 1.9%), but absent in human clinical strains. All the STEC O145 strains of bovine fecal, hide, and human origin carried intimin (*eae*) subtype γ. They also carried LEE-encoded type III secretory system proteins such as *tir*, *espA*, *espB*, and *espF*. All the strains carried *tir*, *espA*, and *espB*, whereas *espF* was present at a frequency of 93.8%, 92.5%, and 100% in cattle fecal, hide and human clinical strains, respectively. Apart from intimin, they also carried other adhesins such as *iha* (IrgA homologue adhesin). The non-LEE encoded effector protein encoding genes such as *nleA*, *nleB*, and *nleC* were present in all the strains. Additionally, they also carried phage-encoded type III secretory system protein encoding genes such as *espI*, *espJ*, *cif*, and *tccP*. All the strains carried *espJ* (except one human strain) and *cif* (except one human strain), *espI* was present only in five strains isolated

**Table 1. Average draft genome size and average number of different categories of genes in STEC O145 strains (n = 71) isolated from human and cattle sources based on RAST subsystem annotation.**

| Source | Draft genome size (Mb) | Functional categories of genes, Mean (Range) | | | | |
|---|---|---|---|---|---|---|
| | | Virulence, disease, and defense | Phages, prophages, transposable elements and plasmids | Membrane transport | Iron acquisition and metabolism | Stress response |
| Cattle feces (n = 16) | 5.41 (5.25–5.63) | 112 (110–118) | 260 (224–291) | 174 (154–199) | 75 (74–75) | 189 (185–191) |
| Cattle hide (n = 53) | 5.28 (5.21–5.46) | 111 (110–115) | 238 (203–268) | 179 (154–188) | 75 (74–75) | 190 (183–192) |
| Human clinical (n = 2) | 5.29 (5.24–5.33) | 111 (110–111) | 259 (252–266) | 170 (160–179) | 75 | 190 (187–192) |

**Table 2. Distribution of virulence genes in STEC O145 strains from cattle and human sources (n = 71).**

| Virulence genes | Product | Source of *E. coli* O145 strains | | |
|---|---|---|---|---|
| | | Cattle feces (n = 16) | Cattle hide (n = 53) | Human (n = 2) |
| **Shiga toxins** | | No. of strains positive | | |
| *stx*1a | Shiga toxin 1 subtype a | 10 | 51 | 2 |
| *stx*2a | Shiga toxin 2 subtype a | 5 | 0 | 1 |
| *stx*2c | Shiga toxin 2 subtype c | 1 | 1 | 0 |
| **Adhesins** | | | | |
| *eae* | Intimin | 16 | 53 | 2 |
| *iha* | IrgA homologue adhesin | 16 | 52 | 2 |
| **LEE encoded Type III secretory system proteins** | | | | |
| *tir* | Translocated intimin receptor | 16 | 53 | 2 |
| *espA* | EPEC secreted protein A | 16 | 53 | 2 |
| *espB* | EPEC secreted protein B | 16 | 53 | 2 |
| *espF* | EPEC secreted protein F | 15 | 49 | 2 |
| **Non-LEE encoded effector proteins** | | | | |
| *nleA* | Non-LEE encoded effector protein A | 16 | 53 | 2 |
| *nleB* | Non-LEE encoded effector protein B | 16 | 53 | 2 |
| *nleC* | Non-LEE encoded effector protein C | 16 | 53 | 2 |
| **Phage encoded type III secretory system proteins** | | | | |
| *espI* | *E. coli*-secreted protein I | 5 | 0 | 0 |
| *espJ* | *E. coli*-secreted protein J | 16 | 53 | 1 |
| *cif* | Cell-cycle inhibiting factor | 16 | 53 | 1 |
| *tccP* | Tir-cytoskeleton coupling protein | 15 | 44 | 2 |
| **Plasmid encoded virulence factors** | | | | |
| *ehxA* | Enterohemolysin | 14 | 38 | 1 |
| *katP* | Catalase peroxidase | 10 | 40 | 1 |
| *espP* | Extracellular serine protease | 16 | 47 | 1 |
| **Antimicrobial resistance genes** | | | | |
| *tetA* | Tetracycline resistance | 1 | 1 | 0 |
| *tetB* | Tetracycline resistance | 1 | 0 | 0 |
| *strA* | Aminoglycoside resistance | 1 | 1 | 0 |
| *strB* | Aminoglycoside resistance | 1 | 1 | 0 |
| *sul2* | Sulphonamide resistance | 2 | 1 | 0 |
| *floR* | Phenicol resistance | 1 | 1 | 0 |
| *bla*$_{CMY2}$ | Beta-lactam resistance | 1 | 1 | 0 |
| **Other** | | | | |
| *iss* | Increased serum survival | 16 | 53 | 2 |
| *cba* | Colicin B | 0 | 1 | 0 |
| *astA* | EAST-1 heat-stable toxin | 16 | 53 | 2 |

from cattle feces, *tccP* was present at a frequency of 93.8%, 83%, and 100% in cattle fecal, hide and human clinical strains. Plasmid-encoded virulence genes such as *ehxA* (87.5% of cattle fecal, 71.7% of cattle hide, and 50% of human clinical strains), *katP* (62.5% of cattle fecal, 75.5% of cattle hide, and 50% of human clinical strains), and *espP* (100% of cattle fecal, 88.7% of cattle hide, and 50% of human clinical strains) were also present in STEC O145 strains. The gene encoding EAST-1 heat-stable toxin (*astA*) was present in all the strains. The virulence gene content of cattle fecal, hide and human strains is provided in Table 2. *Escherichia coli* O145:H28 strain RM12581 carried virulence genes such as *stx*2a, *astA*, LEE-encoded type III

secretory system protein genes (*eae*, *tir*, *espA*, *espB*), non-LEE encoded genes (*nleA*, *nleB*, *nleC*), and phage-encoded type III secretory system protein encoding genes (*espI*, *espJ*, *cif*, and *tccP*).

## Antimicrobial resistance genes

Only a few strains exhibited antimicrobial resistance genes. Two fecal and one hide strains carried antimicrobial resistance genes to aminoglycosides, tetracyclines, sulfonamides, phenicols, and β-lactams. Aminoglycoside resistance genes (*strA* and *strB*) were found in one fecal and one hide strains. Tetracyclin resistance genes, *tetA*, was carried by two strains (one fecal and one hide strain), and *tetB* was carried by one fecal strain. Sulfonamide resistance gene (*sul2*) was present in two fecal and one hide strain. Phenicol resistance gene (*floR*) was carried by one fecal and one hide strain. Beta-lactamase resistance gene ($bla_{CMY-2}$) was carried by one fecal and one hide strains (Table 2).

## Plasmid and prophage sequences

The most common plasmid sequences found in STEC O145 strains were IncFIB (16 cattle fecal, 48 cattle hide, and one human strains) and IncB/O/K/Z (16 cattle fecal, 51 cattle hide, and one human strains). Other plasmid sequences found were IncI2 (one cattle fecal), IncA/C2 (one cattle fecal and one cattle hide), pO111 (one cattle fecal), IncH12 (one cattle fecal) and IncH12A (one cattle fecal) (Table 3). *Escherichia coli* O145:H28 strain RM12581 carried IncFIB, IncB/O/K/Z and IncA/C2.

Average number of phage sequences were 16.7 (11–22) and 14.9 (9–20) in cattle fecal and hide strains, respectively. Average number of intact, incomplete and questionable phage sequences based on PHASTER scores in cattle and human strains is provided in Table 4. Most common phage sequences found among cattle and human sources were *E. coli* bacteriophage WPhi (100% of cattle fecal and human clinical strains, 96.2% of cattle hide strains), *Enterobacteria* phage BP-4795 (100% of cattle fecal and human clinical strains, and 98.1% of cattle hide strains), *Enterobacteria* phage mEP460 (100% of cattle hide and human clinical strains, and 87.5% of cattle fecal strains), *Enterobacteria* phage P88 (100% of human clinical strains, and 87.5% of cattle fecal, and 96.2% of cattle hide strains), *Enterobacteria* phage BP-4795 (100% of cattle fecal and human clinical strains, and 98.1% of cattle hide strains), and *Enterobacteria* phage lambda (100% of human clinical strains, 81.3% of cattle fecal, and 69.8% cattle hide strains) (Table 5). *Escherichia coli* O145:H28 strain RM12581 carried *Enterobacteria* phages such as lambda, BP_4795, YYZ-2008, cdtI, P88, P4, VT2 phi_272, *E. coli* phage WPhi, *Salmonella* phage SEN34, *stx*2 converting phage 1717.

## CRISPR-Cas system

The subtype of CRISPR-Cas system present in all the bovine and human strains were I-A and I-E. All the strains carried Cas proteins such as Cas3, Csa3, Cas8e, DEDDH, Cse2gr11, Cas5, Cas6e, Cas7, Cas1, Cas2, except two hide strains (2015-10-213G1 and 2015-10-218E) which

**Table 3. Plasmid sequences in STEC O145 strains (n = 71) isolated from human and cattle sources identified by PlasmidFinder 1.3.**

| Source | IncFIB | IncB/O/K/Z | IncI2 | IncA/C2 | pO111 | IncH12 | IncH12A |
|---|---|---|---|---|---|---|---|
| **Cattle feces (n = 16)** | 16 | 16 | 1 | 1 | 1 | 1 | 1 |
| **Cattle hide (n = 53)** | 48 | 51 | 0 | 1 | 0 | 0 | 0 |
| **Human clinical (n = 2)** | 1 | 1 | 0 | 0 | 0 | 0 | 0 |

**Table 4. Total number of phage sequences in STEC O145 strains (n = 71) isolated from human and cattle sources based on PHASTER.**

| Source | Type of phage sequences, Mean (Range)[a] | | | |
|---|---|---|---|---|
| | Total | Intact | Incomplete | Questionable |
| **Cattle feces (n = 16)** | 16.7 (11–22) | 8.2 (6–11) | 6.2 (0–14) | 2.3 (1–4) |
| **Cattle hide (n = 53)** | 14.9 (9–20) | 7.6 (4–10) | 6.1 (3–9) | 1.2 (0–4) |
| **Human clinical (n = 2)** | 15.5 (15–16) | 7 (6–8) | 7 (6–8) | 1.5 (1–2) |

[a]Phage sequences were classified as intact, questionable and incomplete based on the PHASTER scores >90, 70–90, <70, respectively

lacked Cse2gr11, Cas7, and Cas5. One fecal strain (2013-3-109C) lacked Cas7 and Cse2gr11. Characteristic features of CRISPR-cas system of all the strains are provided in Table 6.

## Phylogenetic relationship and sequence types

Phylogenetic analysis of the strains revealed that all the strains carrying the same Shiga toxin subtype clustered together irrespective of the source of isolation. *Escherichia coli* O145:H28 reference sequence (GenBank accession no. CP007136), a strain isolated from romaine lettuce from a multistate outbreak of *E. coli* O145 infections in the United States (carrying *stx*2a) [10], clustered with the cattle fecal strains carrying *stx*2a (Fig 1). However, few cattle fecal strains carrying *stx*1a clustered separately from the hide strains carrying *stx*1a. Two human clinical

**Table 5. Intact prophage sequences present in *E. coli* O145 strains of cattle and human origin (n = 71).**

| Prophage | Cattle feces (n = 16) | Cattle hide (n = 53) | Human clinical (n = 2) |
|---|---|---|---|
| *Escherichia coli* **bacteriophage WPhi** | 16 | 51 | 2 |
| *Escherichia* **phage pro147** | 0 | 2 | 0 |
| *Escherichia* **phage pro483** | 0 | 2 | 0 |
| *Enterobacteria* **phage mEp460** | 14 | 53 | 2 |
| *Enterobacteria* **phage P88** | 14 | 51 | 2 |
| *Enterobacteria* **phage cdtI** | 2 | 0 | 0 |
| *Enterobacteria* **phage Lambda** | 13 | 37 | 2 |
| *Enterobacteria* **phage BP-4795** | 16 | 52 | 2 |
| *Enterobacteria* **phage P4** | 5 | 9 | 0 |
| *Enterobacteria* **phage 933W** | 5 | 0 | 0 |
| *Enterobacteria* **phage HK630** | 4 | 18 | 0 |
| *Enterobacteria* **phage phi27** | 1 | 0 | 0 |
| *Enterobacteria* **phage YYZ-2008** | 1 | 2 | 0 |
| *Enterobacteria phage* **mEP043 C-1** | 2 | 0 | 0 |
| *Enterobacteria* **phage Min27** | 1 | 0 | 0 |
| *Enterobacteria* **phage P1** | 1 | 1 | 0 |
| *Enterobacteria* **phage UAB_Phi20** | 0 | 1 | 0 |
| *Enterobacteria* **phage Sf6** | 0 | 1 | 0 |
| *Enterobacteria* **phage Mu** | 0 | 2 | 0 |
| *Shigella* **phage SfII** | 1 | 2 | 0 |
| *stx*2 **converting phage 1717** | 3 | 4 | 0 |
| *stx*2 **converting phage 86** | 1 | 0 | 0 |
| **Phage Gifsy-1** | 1 | 0 | 0 |
| *Flavobacterium* **phage 1H** | 0 | 1 | 0 |
| *Erwinia* **phage vB_EamM_Caitlin** | 0 | 2 | 0 |

**Table 6. Characteristic features of CRISPR-cas system in *E. coli* O145 strains isolated from cattle and human strains based on annotation by CRISPRone tool.**

| Source | Subtype | Cas proteins | Average no. of repeats (Range) | Average length of repeats (Range) | Average no. of spacers (Range) | Average length of spacers (Range) | No. of strains carrying questionable CRISPR* |
|---|---|---|---|---|---|---|---|
| **Cattle feces (n = 16)** | I-E, I-A | Cas1, Cas2, Cas3, Csa3, Cas5, Cas 6e, Cas7, Cas8e, Cse2gr11, DEDDH | 8 (3–12) | 29 (28–30) | 6 (2–10) | 32 (31–33) | 11 |
| **Cattle hide (n = 53)** | I-E, I-A | Cas1, Cas2, Cas3, Csa3, Cas5, Cas 6e, Cas7, Cas8e, Cse2gr11, DEDDH | 5 (3–11) | 29 (28–30) | 4 (2–9) | 32 (32–33) | 21 |
| **Human clinical (n = 2)** | I-E, I-A | Cas1, Cas2, Cas3, Csa3, Cas5, Cas 6e, Cas7, Cas8e, Cse2gr11, DEDDH | 3 | 2 | 29 | 32 | 2 |

* A sequence is considered to contain a questionable CRISPR–Cas system if CRISPR array(s) are predicted, but no cas genes are found in the sequence

strains (one carrying *stx*1a, one carrying both *stx*1a and *stx*2a) also clustered with hide strains carrying *stx*1a. All *E. coli* O145 strains of cattle and human origin belonged to ST-32.

## Discussion

Understanding the virulence potential of STEC O145 strains isolated from cattle feces and hide samples is useful in estimating the risk associated with different sources of human illnesses. The whole genome sequences of STEC O145 strains isolated from cattle feces, hide and human clinical strains were analyzed to determine their virulence potential. All strains carried flagellar type H28, which is the most common flagellar type carried by human outbreak and environmental O145 strains [10, 11]. The average draft genome size of cattle fecal strains was larger than cattle hide and human clinical strains, which appears to be because of the higher average number of genes associated with mobile genetic elements in cattle fecal strains than cattle hide and human clinical strains. This suggests that the size of the genomes were proportional to the mobile genetic elements. Similar results have been reported in previous studies [26].

The virulence gene profile of STEC O145 strains isolated from cattle feces, hide, and human clinical strains were similar. They carried Shiga toxins, LEE-encoded type III secretory system proteins, and plasmid-encoded virulence genes. The environmental and outbreak *E. coli* O145 strains were also found to carry core EHEC virulence determinants [11]. A majority of cattle strains, both fecal and hide (61/69), carried *stx*1 gene and Shiga toxin 1a was the only subtype present in all strains. Shiga toxin 1a is the most common subtype of *stx*1 found in non-O157 STEC serogroups of cattle [27, 28]. Shiga toxin 2a (*stx*2a) and *stx*2c were the subtypes of *stx*2 found in *E. coli* O145 strains of cattle and human origin. Similar findings were also reported in environmental *E. coli* O145 strains by Carter et al. [11]. Shiga toxin 2a (*stx*2a) and *stx*2c carrying STEC strains have been previously reported to be frequently associated with HUS in humans [29–31]. Shiga toxin 2a was also carried by *E. coli* O145 strains isolated from ice-cream associated outbreak in Belgium and lettuce associated outbreak in the United States [10]. In our study, a majority of the fecal strains positive for *stx*2 belonged to subtype 2a (31.3%), and surprizingly, none of the hide strains contained subtype 2a. Shiga toxin 2a (*stx*2a) and *stx*2c were the most prevalent subtypes of *stx*2 in non-O157 STEC serogroups isolated from cattle [27, 28]. However, none of our strains carried *stx*2d, which is another most prevalent *stx*2 subtype reported in non-O157 STEC serogroups [27, 28]. One strain isolated from cattle hide was negative for *stx*, although, it tested positive for *stx*2 by end-point PCR, which may likely be due to the loss of *stx*-encoding phage. *Escherichia coli*

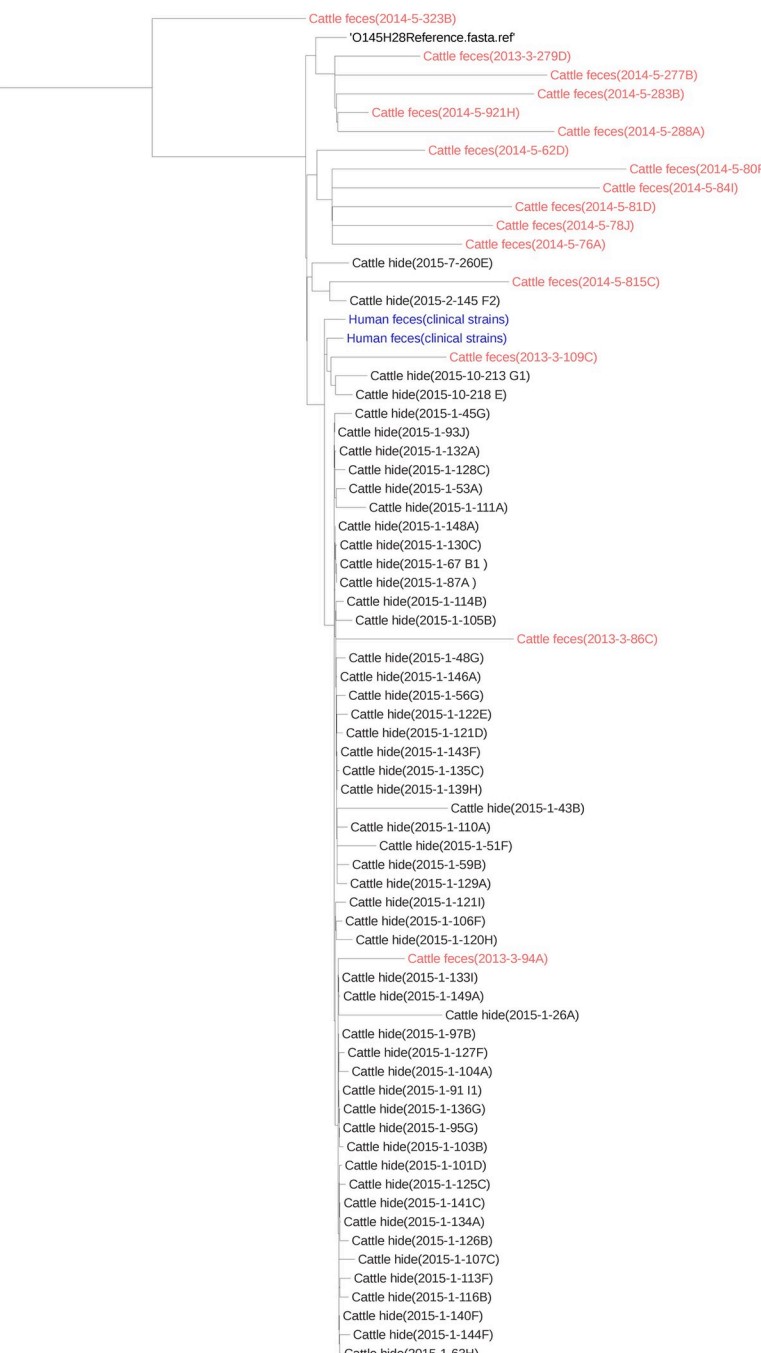

**Fig 1. Phylogenetic tree of *E. coli* O145 strains isolated from cattle feces, hide and human clinical cases using Parsnp v1.2 and visualized using FigTree 1.4.3.** Parsnp uses compressed suffix graph (CSG) to identify maximal unique matches (MUM). A divide and conquer algorithm further refined the MUMs, and locally collinear blocks (LCB) of MUMs were identified which formed the basis of core genome alignment [25].

O145 strains from all three sources carried genes associated with LEE and phage-encoded type III secretory system and *nle* genes. Locus of enterocyte effacement-encoded type III secretory system proteins are involved in the formation of attachment and effacement lesions in host epithelial cells [32, 33]. Genes encoding non-LEE encoded effectors have been reported to be associated with complications of STEC infections such as HUS [34, 35]. Intimin subtype γ was the only subtype of intimin found among all the strains. Intimin gamma subtype was found to be frequently associated with *E. coli* O145, O55, and O157 strains isolated from cattle and human sources [36]. Strains from all three sources were also positive for plasmid-encoded (pO157) virulence genes such as *katP*, *ehxA*, and *espP*. Whole genome sequence-based analysis of *E. coli* O145 outbreak strains revealed that the virulence genes carried by pO145 were similar to that carried by pO157, although they lacked *katP* [10]. Plasmid-encoded virulence genes have been reported to be involved in the pathogenesis of STEC [37–39]. In our study, pO157 plasmid sequence was not found in O145 strains, although they carried virulence genes encoded by pO157. *Escherichia coli* heat-stable enterotoxin 1 (EAST1), encoded by *astA* was present in *E. coli* O145 strains isolated from all three sources. This gene was reported to be frequently associated with diarrhea caused by typical and atypical enteropathogenic *E. coli* [40].

Mobile genetic elements have been shown to play a major role in genomic diversity and evolution of pathogenic *E. coli* [41]. A total of seven types of plasmid sequences were found in *E. coli* O145 strains of cattle and human origin. IncFIB was the most frequently found plasmid sequence in cattle fecal, hide and human strains. IncF plasmid has been most frequently found in *E. coli* strains carrying antibiotic resistance genes such as *tet*(A), $bla_{TEM-1}$, and $bla_{CTX-M-15}$ [42, 43]. A majority of the virulence associated plasmids in *E. coli* belong to IncF incompatibility family [44]. Another most commonly identified plasmid sequence among STEC O145 strains was IncB/O/K/Z. IncB/O/K/Z carrying genes encoding penicillin resistance ($bla_{TEM}$) was found in *Shigella* strains associated with outbreaks [45]. IncA/C2 plasmid sequence was present in one each of cattle fecal and hide strains. IncA/C2 plasmid carrying antibiotic resistance genes were found in *E. coli* O145 strains associated with multistate outbreak in the US in 2010 [46]. *Escherichia coli* O145:H28 human outbreak strain RM12581 used as control in our study also carried IncFIB, IncB/O/K/Z and IncA/C2 plasmid sequences. However, only few strains carried antimicrobial resistance genes (3/71; two cattle fecal and one hide strains). They carried genes encoding resistance for tetracycline (*tetA* and *tetB*), aminoglycoside (*strA* and *strB*), sulphonamide (*sul2*), phenicol (*floR*) and beta-lactam ($bla_{CMY-2}$). Antimicrobial resistance genes, such as *floR*, *strA*, *strB*, *sul2*, and *tetA*, were found in *E. coli* O145 strains associated with multistate outbreak in the US in 2010 [46].

Bacteriophages also encode important virulence factors such as Shiga toxins. In our study, we identified a total of 25 types of prophages, of which seven of them were lambdoid phages. Lambdoid phages have been shown to be predominant in the genomes of enterohemorrhagic *E. coli*, and are shown to carry type III secretory system effector protein encoding genes [41, 47]. Lamdoid prophages (*Enterobacteria* phage BP-4795, *Enterobacteria* phage mEP460, *Enterobacteria* phage lambda, *Enterobacteria* phage phi27, *Enterobacteria* phage HK630, Stx2-converting phage 1717, *Enterobacteria* phage cdtI, *Enterobacteria* phage mEP043 C-1) found in *E. coli* O145 strains in our study were also previously reported in *E. coli* O145 human outbreak strains [10, 26]

Phylogenetic analysis of core genomes indicated that the bovine STEC O145 strains carrying the same Shiga toxin subtype clustered together, suggesting that the acquisition of specific subtype of Shiga toxin might be influenced by the genetic background of the STEC strain. However, many of the cattle fecal strains clustered separately from the hide strains, although

a few (4 strains) clustered with the hide strains. Because the fecal isolates and hide isolates were from different cattle and from different feedlots, a direct comparison between the two groups cannot be made. A great deal of genetic diversity in the clonal population of environmental STEC O145 strains (cattle, feral pig, sediment, water, wildlifw, and human) have been reported [11]. Further studies on the in-depth analyses of the core genomes are required to determine the reasons for genetic diversity of these strains. A limitation of our study was that it included only a few fecal isolates (n = 16) collected from multiple feedlots, and more importantly, the fecal and hide strains were not from the same cattle. Obviously, including more O145 strains from different sources, particularly strains that are matched by source, may provide a more comprehensive analysis of genetic diversity of the cattle O145 strains. All O145 strains of cattle and human origin belonged to ST-32. *Escherichia coli* O145 human outbreak strain RM12581 used as control also belonged to ST-32. *Escherichia coli* O145 strains isolated from wildlife and human outbreak strain (RM13514) belonging to ST-32 has been previously reported [11, 26].

In conclusion, STEC O145 strains isolated from cattle feces and hide samples carried the same flagellar type (H28) and their virulence gene profile was similar. All the strains belonged to the same sequence type (ST-32). Additionally, human clinical strains clustered closely with cattle hide strains. Our study demonstrated the presence of potentially pathogenic STEC O145 strains in cattle feces and hide, which could cause foodborne illness in humans.

## Acknowledgments

This is contribution no. 19-134-J from the Kansas Agricultural Experiment Station, Manhattan.

## Author Contributions

**Conceptualization:** Jianfa Bai, Jianghong Meng, T. G. Nagaraja.

**Formal analysis:** Jay N. Worley, Xin Gao.

**Investigation:** Pragathi B. Shridhar, Xun Yang, Lance W. Noll, Xiaorong Shi.

**Writing – original draft:** Pragathi B. Shridhar.

**Writing – review & editing:** Jianfa Bai, Jianghong Meng, T. G. Nagaraja.

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
