## [Decision Letter · Decision Letter 0]

17 Sep 2019

PONE-D-19-20091

Analysis of Virulence Potential of Escherichia coli O145 Isolated from Cattle Feces and Hide Samples Based on Whole Genome Sequencing

PLOS ONE

Dear Dr. Nagaraja,

Thank you for submitting your manuscript to PLOS ONE. After careful consideration, we feel that it has merit but does not fully meet PLOS ONE’s publication criteria as it currently stands. Therefore, we invite you to submit a revised version of the manuscript that addresses the points raised during the review process.

I have received the reviews of your manuscript. While your paper addresses an interesting question, the reviewers stated several concerns about your study that need to be addressed.  Please see reviewers' insightful comments below and 

We would appreciate receiving your revised manuscript by Nov 01 2019 11:59PM. To enhance the reproducibility of your results, we recommend that if applicable you deposit your laboratory protocols in protocols.io, where a protocol can be assigned its own identifier (DOI) such that it can be cited independently in the future. For instructions see: http://journals.plos.org/plosone/s/submission-guidelines#loc-laboratory-protocols

We look forward to receiving your revised manuscript.

Kind regards,

Baochuan Lin, Ph.D.

Academic Editor

PLOS ONE

Journal Requirements:

Reviewers' comments:

Reviewer's Responses to Questions

**Comments to the Author**

1. Is the manuscript technically sound, and do the data support the conclusions?

Reviewer #1: No

Reviewer #2: Yes

2. Has the statistical analysis been performed appropriately and rigorously? 

Reviewer #1: Yes

Reviewer #2: Yes

3. Have the authors made all data underlying the findings in their manuscript fully available?

Reviewer #1: Yes

Reviewer #2: Yes

4. Is the manuscript presented in an intelligible fashion and written in standard English?

Reviewer #1: Yes

Reviewer #2: Yes

5. Review Comments to the Author

Reviewer #1: Comments.

Author line: No affiliation for number 3 (Lance W. Noll3)

Abstract and elsewhere in the manuscript: the “average genome size” should be changed to say the “approximate average genome size” as these are just draft genomes.

Introduction, lines 58 and 63. Remove “have” from the sentence.

Materials and Methods, lines 82-85. I was surprise to see such few isolates used to make a major claim of cattle STEC O145:H28 isolates potential can cause human illness. There were only two human isolates used in this study. I don’t know how there can be any power to your statistical analysis with just two strains. I am not even sure the 16 cattle feces isolates would be enough to represent all the genetic diversity of STEC O145:H28 in cattle. There are sequenced STEC O145 that can be downloaded from Genbank to increase the number of human isolates and increase the diversity of the isolates used in this study.

Materials and Methods, lines 82-85. The two references sited (14 and 15) are both from cattle feces. I didn’t see where any of them were from an abattoir. How did you select which cattle fecal isolates to sequence? Between the two papers in the reference, there should have been well over 100 isolates you could sequence. Were the hide samples from the abattoir from the same or feed yard? They are closely related which makes me think they are epidemiologically related. How do you explain the clustering of cattle feces and hide isolates on the tree when the hide isolate more than likely came from feces?

Discussion, line 298. Remove “the” before ”fecal strains …”.

Reviewer #2: The manuscript describes the characterization of Shiga toxin-producing Escherichia coli O145 serogroup strains isolated from cattle feces, hide and human clinical samples. The paper is of interest, particularly due to the lack of information from the O145 serogroup, but some aspects need to be clarified.

1. I suggest using “big six non-O157 STEC” instead of “top-7 STEC”.

2. I suggest mention that your sample size is small (in the conclusion of abstract and discussion)

3. Lines 45 and 46 are not suitable for initiating of introduction, I suggest removing them.

4. Line 69: please mention the time of sampling for O145 collection? How many years? How many months? Since what year and month to what year and month?

5. Line 86 data is more appropriate for results section not for materials and methods.

9. Line 99: Correct the “))”

10. Line 144: "[260 (224-291)]" seems to be better than "(260 [224-291])". Check it.

11. Because of the small sample size, I disagree with percentage in the text and table 2. Consult with an epidemiologist.

12. in figure captions: What algorithm is used? (For example: UPGMA), what similarity coefficient is used? (For example: Dice)

13. In discussion, you can compare your stx subtypes with some other studies to show the importance of cattle as a disseminator of the subtypes:

For example: Virulence genes, Shiga toxin subtypes, major O-serogroups, and phylogenetic background of Shiga toxin-producing Escherichia coli strains isolated from cattle in Iran. Microbial pathogenesis. 2017 Aug 1;109:274-9.

6. PLOS authors have the option to publish the peer review history of their article (what does this mean?). If published, this will include your full peer review and any attached files.

Reviewer #1: No

Reviewer #2: No

---

## [Author Response · Author response to Decision Letter 0]

3 Oct 2019

I have uploaded a response document that addresses the comments

---

## [Decision Letter · Decision Letter 1]

29 Oct 2019

Analysis of Virulence Potential of Escherichia coli O145 Isolated from Cattle Feces and Hide Samples Based on Whole Genome Sequencing

PONE-D-19-20091R1

Dear Dr. Nagaraja,

We are pleased to inform you that your manuscript has been judged scientifically suitable for publication and will be formally accepted for publication once it complies with all outstanding technical requirements.

With kind regards,

Baochuan Lin, Ph.D.

Academic Editor

PLOS ONE

Additional Editor Comments (optional):

Reviewers' comments:

Reviewer's Responses to Questions

**Comments to the Author**

1. If the authors have adequately addressed your comments raised in a previous round of review and you feel that this manuscript is now acceptable for publication, you may indicate that here to bypass the “Comments to the Author” section, enter your conflict of interest statement in the “Confidential to Editor” section, and submit your "Accept" recommendation.

Reviewer #1: All comments have been addressed

Reviewer #2: All comments have been addressed

2. Is the manuscript technically sound, and do the data support the conclusions?

Reviewer #1: (No Response)

Reviewer #2: Yes

3. Has the statistical analysis been performed appropriately and rigorously? 

Reviewer #1: (No Response)

Reviewer #2: Yes

4. Have the authors made all data underlying the findings in their manuscript fully available?

Reviewer #1: (No Response)

Reviewer #2: Yes

5. Is the manuscript presented in an intelligible fashion and written in standard English?

Reviewer #1: (No Response)

Reviewer #2: Yes

6. Review Comments to the Author

Reviewer #1: (No Response)

Reviewer #2: (No Response)

7. PLOS authors have the option to publish the peer review history of their article (what does this mean?). If published, this will include your full peer review and any attached files.

Reviewer #1: No

Reviewer #2: No

---

## [Editor Report · Acceptance letter]

19 Nov 2019

PONE-D-19-20091R1 

Analysis of Virulence Potential of Escherichia coli O145 Isolated from Cattle Feces and Hide Samples Based on Whole Genome Sequencing 

Dear Dr. Nagaraja:

I am pleased to inform you that your manuscript has been deemed suitable for publication in PLOS ONE. Congratulations! Your manuscript is now with our production department. 

With kind regards,

on behalf of

Dr. Baochuan Lin 

Academic Editor

PLOS ONE